# Analysing microbiome intervention design studies: Comparison of alternative multivariate statistical methods

**Maryia Khomich**[1,2]*, **Ingrid Måge**[3]*, **Ida Rud**[1], **Ingunn Berget**[3]

**1** Division of Food Science, Department of Food Safety and Quality, Nofima – Norwegian Institute of Food, Fisheries and Aquaculture Research, Ås, Norway, **2** Department of Clinical Science, University of Bergen, Bergen, Norway, **3** Division of Food Science, Department of Raw Materials and Process Optimisation, Nofima – Norwegian Institute of Food, Fisheries and Aquaculture Research, Ås, Norway

* marykhomich@gmail.com, maryia.khomich@uib.no (MK); ingrid.mage@nofima.no (IM)

**Data Availability Statement:** All relevant data are within the paper and its Supporting information files.

**Funding:** This work was funded by Nofima – Norwegian Institute of Food, Fisheries and

## Abstract

The diet plays a major role in shaping gut microbiome composition and function in both humans and animals, and dietary intervention trials are often used to investigate and understand these effects. A plethora of statistical methods for analysing the differential abundance of microbial taxa exists, and new methods are constantly being developed, but there is a lack of benchmarking studies and clear consensus on the best multivariate statistical practices. This makes it hard for a biologist to decide which method to use. We compared the outcomes of generic multivariate ANOVA (ASCA and FFMANOVA) against statistical methods commonly used for community analyses (PERMANOVA and SIMPER) and methods designed for analysis of count data from high-throughput sequencing experiments (ALDEx2, ANCOM and DESeq2). The comparison is based on both simulated data and five published dietary intervention trials representing different subjects and study designs. We found that the methods testing differences at the community level were in agreement regarding both effect size and statistical significance. However, the methods that provided ranking and identification of differentially abundant operational taxonomic units (OTUs) gave incongruent results, implying that the choice of method is likely to influence the biological interpretations. The generic multivariate ANOVA tools have the flexibility needed for analysing multifactorial experiments and provide outputs at both the community and OTU levels; good performance in the simulation studies suggests that these statistical tools are also suitable for microbiome data sets.

## Introduction

The microbiome has emerged as an important link to health and disease [1]. Microbiome analysis methods are rapidly advancing, in particular in areas such as compositional data analysis, multi-omics and data integration [2, 3]. A clear understanding of the type of data being analysed is crucial, given the growing number of studies uncovering the key role of microbiome, its composition and functions following diet intervention or medical treatment [4]. At present, analysis of complex microbial data benefits from adapting the multivariate statistical

Aquaculture Research and Foundation for Research Levy on Agricultural Products (Research Council of Norway projects No. 262306, 262308, 314111 and 314743). The funding sponsors had no role in the design of the study, the collection, analyses and interpretation of data, in the writing of the manuscript or in the decision to distribute the results.

**Competing interests:** The authors declare no competing interests.

**Abbreviations:** ALDEx2, ANOVA-like differential expression tool for high-throughput sequencing data; ANCOM, Analysis of composition of microbiomes; ANOSIM, Analysis of similarities; ANOVA, Analysis of variance; ASCA, ANOVA-simultaneous component analysis; DESeq2, Differential gene expression analysis based on the negative binomial distribution; FDR, False discovery rate; FFMANOVA, Fifty-fifty multivariate ANOVA; OTU, Operational taxonomic unit; PERMANOVA, Permutational multivariate analysis of variance; PLS-DA, Partial least squares discriminant analysis; SIMPER, Similarity percentage.

toolbox from ecology and environmental sciences, and a proper choice of statistical tools is becoming increasingly important [5–7]. However, a lack of benchmarking studies and clear consensus on the best multivariate statistical practices make comparisons across microbiome data sets difficult [2, 8]. New methods are often tested by simulation studies, but there is always a concern that simulations can be biased towards the tested statistical model and cannot mimic the complexity of real microbiome data [6, 9]. Moreover, newly introduced tools are often optimised, whereas the comparison of several statistical methods implies the use of standard or default parameters [6, 9]. It is therefore of interest to compare existing methods on real data sets of different complexity, in addition to simulation studies, to better understand how choice of method affects the results.

Different statistical methods have different properties, and the choice of method should depend on the scientific question, experimental design, data characteristics and expected relationships among the variables. Furthermore, the choice of method is often biased by the research groups' tradition and familiarity with specific "toolboxes". Main differences between existing statistical approaches for analysing microbiome data are related to: (1) explorative versus confirmative; (2) univariate versus multivariate; (3) parametric versus nonparametric; (4) linear versus nonlinear; (5) compositional versus non-compositional; (6) distance-based versus count/abundance-based; and (7) incorporating phylogenetic information into the analysis or not [10–12]. Here, we explore statistical methods for analysing microbiome data from designed experiments with a focus on dietary intervention trials. In contrast to observational studies, these are usually small in sample size but performed in (semi)-controlled environments and tailored to a specific research hypothesis. The studies often include multiple experimental factors, possibly with more than two levels, and it is therefore natural to turn to analysis of variance (ANOVA)-like methods. Notably, most published analytical tools in microbiome research are essentially univariate [6], which led us to the conclusion that comparison of alternative multivariate statistical tools is sorely missing. From a biologist's point of view, it is also important that the methods are easy to interpret, both at the multivariate (microbial community) and univariate (microbial taxa or operational taxonomic units, OTUs) levels (see Fig 1).

## Distance-based methods

The distance-based methods are multivariate since multiple variables (microbial OTUs) are used to calculate pairwise distances between samples. Among distance-based methods, permutational multivariate analysis of variance (PERMANOVA) is the most widely used and more powerful than the analysis of similarities (ANOSIM) to detect changes in community structure [13–15]. Both methods may be implemented with any dissimilarity metric. Among abundance-based beta diversity indices, Bray-Curtis is the most common choice for count data [16, 17]. The most widely applied phylogenetic beta diversity indices are UniFrac-type metrics [17–19]. However, UniFrac is unsuitable as a distance metric for studies with a small sample size, which is usually the case for dietary intervention trials [20, 21]. Both PERMANOVA and ANOSIM test differences at the community level but do not provide any information at the OTU level. Similarity percentage analysis (SIMPER) works at the univariate level by computing the relative contribution of each analysed microbial taxon (i.e. OTU) to the overall average Bray-Curtis dissimilarities by pairwise comparison of two or more groups [15]. To the best of our knowledge, no such method exists for the other distance metrics.

Distance-based methods have their strengths and weaknesses that are important to account for beforehand. ANOSIM cannot deal with multifactorial designs, and both ANOSIM and PERMANOVA may have problems detecting differences unless they are present in taxa with

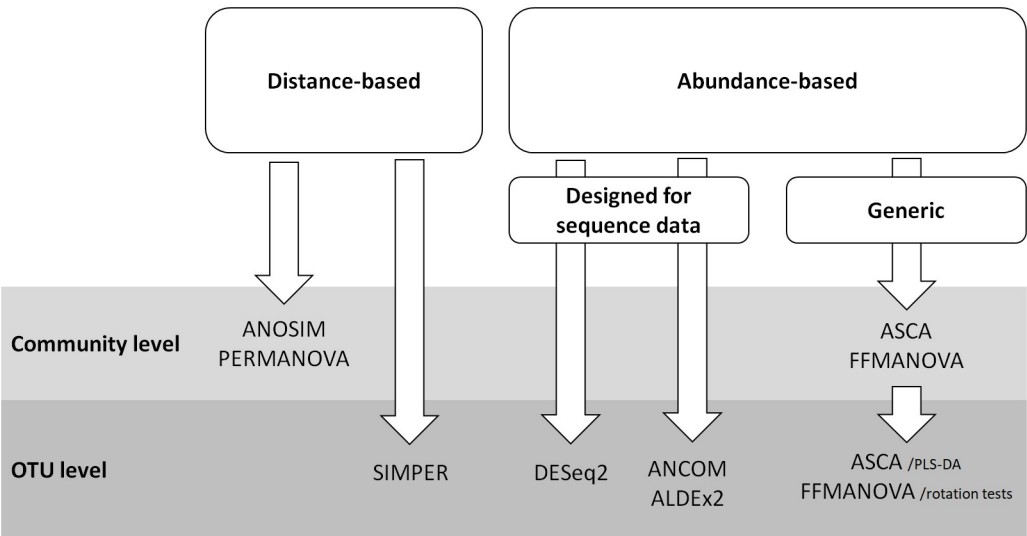

**Fig 1. A diagram of statistical methods used in the study.**

high variability [22]. Newer methods aimed at assigning more interpretable effect sizes are under constant development [6, 23]. For example, the more flexible PERMANOVA-S is an extension to existing distance-based methods that can adjust for covariates and simultaneously incorporate multiple distance metrics [24]. However, these methods do not consider covariance/correlation between microbial species, and they encounter significant power loss if *all* microbial species are used for distance calculations [25].

## Abundance-based methods

The abundance-based methods can be either univariate (analysing each OTU individually) or multivariate (focusing on covariance structure between OTUs). There are two main approaches to deal with the special nature of abundance data: (1) application of methods that consider the distribution of count data or (2) compositional data analysis (CoDa) based on log-ratio transformed count data [26, 27]. Statistical methods designed for high-throughput sequencing data are ANOVA-like differential expression analysis (ALDEx2) [26], analysis of composition of microbiomes (ANCOM) [28], edgeR [29] and DESeq/DESeq2 [30]. edgeR and DESeq2 model count data directly using generalized linear models with the negative binomial distribution and the logistic link, respectively, whereas ALDEx2 and ANCOM use the log-ratio transformation prior to univariate assessment of statistical significance for individual OTUs. DESeq2 and edgeR are based on the same modelling approach but differ in normalisation, outlier handling, and other adjustable parameters; these methods had similar performance in simulation studies [30]. Thus, we decided to include only one of the methods—DESeq2—because differences between DESeq2 and edgeR are at a different conceptual level rather than the other methods discussed. ALDEx2 uses a Dirichlet-multinomial probability distribution to estimate abundances from count data and calculates the false discovery rate (FDR) based on Monte Carlo simulations (see Fig 3 in [26] for details). In ANCOM, the compositional nature of the data is considered by testing the log-ratio for all pairs of OTUs, and then counting the number of tests where the log-ratio is significantly different from zero. This number (W-stat) can be used to obtain a ranking of OTUs most likely to differ between the groups. The newly published ANCOM-BC corrects the bias induced by differences in sampling fractions and

provides p-values and confidence intervals for the differential abundance of each OTU [31]. The FDR and power were shown to be similar for both ANCOM and ANCOM-BC, and therefore we limit the present study to ANCOM.

The drawback of univariate methods is that they treat all taxa as independent variables without considering the covariance between the OTUs. Such methods may fail to detect community-level differences [32]. A classical generalisation of ANOVA to multiple variables (MANOVA) cannot be used when the number of variables exceeds the number of samples, as it suffers from the problem of a singularity of covariance matrices and assumptions that are not fulfilled [33, 34]. Novel statistical ANOVA-like methods include fifty-fifty multivariate analysis of variance (FFMANOVA) [35] and ANOVA-simultaneous component analysis (ASCA) [33]. Both methods are based on principal component analysis (PCA), and they can handle multiple collinear responses. In FFMANOVA, the multivariate effects are estimated by a modified variant of classical MANOVA, and OTU-level p-values are obtained by rotation tests which adjust the p-values for multiple testing [36]. For ASCA, the multivariate effects are calculated from combined sums-of-squares from all OTUs, and significance is assessed by permutation testing. ASCA also provides scores and loadings related to each experimental factor, which can be visualised in the same way as for PCA to better understand covariance patterns within the data. The contribution of each OTU can be quantified by the loadings or by partial least squares discriminant analysis (PLS-DA) for pairwise comparisons. ASCA has recently gained popularity in metabolomics [37–39], and both ASCA and FFMANOVA have successfully been applied to microbiome data [40–44].

Linear discriminant analysis effect size (LEfSe) is a stepwise approach that combines univariate analysis with multivariate discriminant analysis [45]. LEfSe has found wide application in microbiome research due to its easy to-use-and-interpret visualization [46, 47], but it is not adapted to experimental designs with several multilevel factors and is therefore not considered in this study.

## Method comparison

An overview of the different methods compared in this study is given in Fig 1 and Table 1. ANOSIM and PERMANOVA provide results only at the community level, while SIMPER, DESeq2, ANCOM and ALDEx2 report results for single OTUs. ASCA and FFMANOVA are generic methods and the only methods that provide results at both the community and OTU level.

The aim of the method comparison was to investigate how different strategies for statistical modelling affect biological inference. At the community level, methods were compared with respect to effect sizes (expressed as percentage of explained variance) and corresponding p-values. At the OTU level, comparison of methods is complex because some methods provide results for an omnibus test of differences between factor levels (FFMANOVA and ANCOM), whereas the other methods provide ranking for specific pairwise comparisons (ASCA and PLS-DA, SIMPER) or contrasts/model coefficients (ALDEx2 and DESeq2). Even so, a biologist will make inferences based on the output provided by the chosen method, and in this context, it is relevant to compare the ranking statistics although the tests are not the same. In our study, the ranking of OTUs was compared by Spearman's rank correlation and by investigation of scatterplots between the different ranking metrics. For the simulated data, where we know which OTUs are differentially abundant, True Positive Rate (TPR) and True Negative Rate (TNR) were also evaluated.

We focused exclusively on designed experiments, which are usually smaller in sample size and are more controlled in contrast to observational studies. We used five published data sets

**Table 1. An overview of statistical methods and their properties.**

| Method | Method name | Number of experimental factors allowed | Parametric | Multivariate | Univariate | Provides output at community level | Statistics for ranking OTUs | Reference |
|---|---|---|---|---|---|---|---|---|
| ALDEx2 | ANOVA-like differential expression tool for high-throughput sequencing data | any | yes | no | yes | no | p-values or effect sizes | [26] |
| ANCOM | Analysis of composition of microbiomes | main factor + covariates | yes | no | yes | no | W-stat for the main variable | [28] |
| ANOSIM | Analysis of similarities | one | no | yes | no | yes | no | [15] |
| ASCA | ANOVA-simultaneous component analysis | any | yes | yes | no | yes | loadings or PLS-DA regression coefficients | [33] |
| DESeq2 | Differential gene expression analysis based on the negative binomial distribution | any | yes (GLM) | no | yes | no | p-values or effect sizes (coefficients) | [30] |
| FFMANOVA | Fifty-fifty multivariate ANOVA | any | yes | yes | yes (rotation tests) | yes | p-values | [35] |
| PERMANOVA | Permutational multivariate analysis of variance | any | no | yes | no | yes | no | [14] |
| SIMPER | Similarity percentage | two-group comparison | no | yes | no | no | permutation p-values | [15] |

ANOVA—analysis of variance; GLM—generalized linear model; OTU—operational taxonomic unit; PLS-DA—partial least squares discriminant analysis.

as a basis for the comparisons (S1 Table). The following criteria for studies to be included were considered: (1) at least two-factorial experimental design with a minimum of two-factor levels; (2) either human or animal gut microbiome surveys; and (3) a taxonomic assignment at the OTU level reported. Diet is the main factor of interest in all five studies, and we restricted our comparisons to this factor.

The simulated data were based on data set 1 (S1 Table) using the same study design and OTU counts as a starting point. Four different scenarios were simulated to investigate how the methods perform in situations with varying effect sizes and different numbers of differentially abundant OTUs. In all scenarios, one of the diet levels was manipulated to be significantly different from the others, and there was no effect of the second experimental factor (dose). See Methods section for further details.

## Results

### Community level

Four of the methods can be used to test the association between diet and the overall microbiome composition, the distance-based ANOSIM and PERMANOVA, and abundance-based ASCA and FFMANOVA. The results for each of the four simulated scenarios are shown in Fig 2, and the results across real data sets are summarised in Table 2.

**Simulated data.** As expected, the multivariate effect size (explained variance) is lowest (around 5%) for the *"Few-Low"* scenario and highest (30–35%) for the *"Many-High"* scenario. The multivariate effect was significant in 100% of the simulations in three scenarios with the highest effect size. For the *"Few-Low"* scenario, FFMANOVA performed best by detecting the effect in 80% of the data sets. For the *"Many"* simulations explained variance was slightly higher with FFMANOVA than with ASCA and PERMANOVA, whereas for the *"Few-High"* simulations the opposite trend was observed. ANOSIM was less consistent than the other

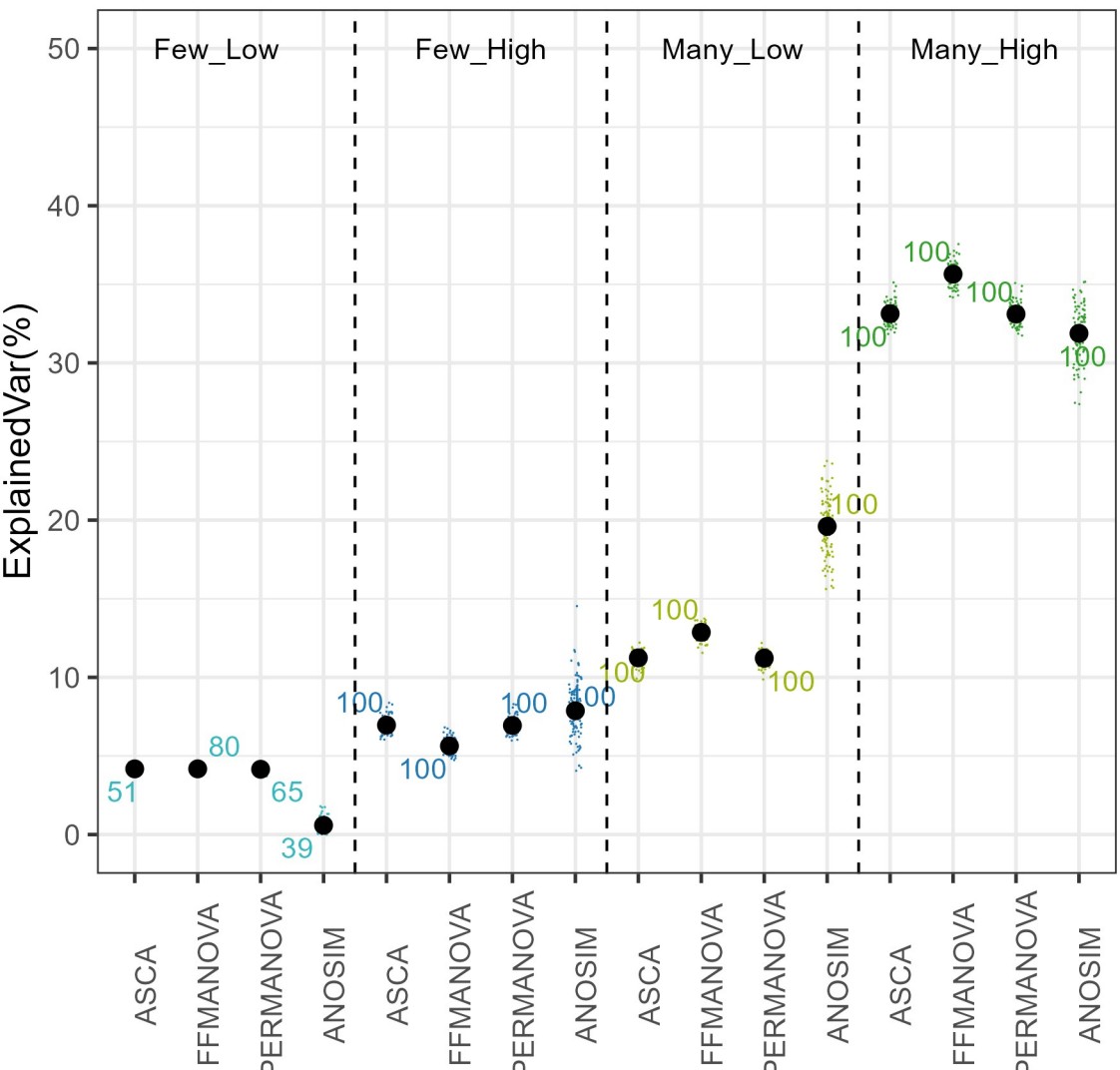

**Fig 2. Explained variance for simulated data and the relative number of simulations where the simulated effect was detected.**
Numbers indicate the percentage of simulated data sets where p-value was significant.

methods, with higher within-scenario variation and higher differences in effect size between the two "*Low*" scenarios.

**Data set 1**. The effect of dietary fibres inulin (IN), cellulose (CE) or brewers spent grain (BSG) on the overall caecal microbiota composition in mice from a study by Moen *et al.* [42] accounted for 34–37% of the explained variance according to the FFMANOVA, ASCA and PERMANOVA. In general, the three methods produced similar results, with slightly smaller p-values by FFMANOVA and ASCA.

**Data set 2**. Lai *et al.* [48] investigated the effect of diet (the main variable), exercise and their interaction on the overall faecal microbiota in sedentary and exercised mice fed high fat or normal fat diet (four groups in total). Similarly, all tested methods, except ANOSIM, produced congruent results (effect of diet 27–31%), with smaller p-values by FFMANOVA and ASCA.

**Table 2. Community-level method comparison across five experimental data sets.**

| Factor/ predictor | FFMANOVA[clr] | | ASCA[clr] | | PERMANOVA[clr] | | ANOSIM[3, clr] | | Factor/ predictor |
|---|---|---|---|---|---|---|---|---|---|
| | Effect size (explained variance), % | p-value[1] | Effect size (explained variance), % | p-value[2] | Effect size (explained variance), % | p-value[2] | Effect size (explained variance), % | p-value[2] | |
| Moen et al., 2016 (data set 1) | | | | | | | Moen et al., 2016 (data set 1) | | |
| Model: OTU ~ fiber*dose | | | | | | | Model: OTU ~ fiber_dose | | |
| fiber | 34.31 | < 0.001 | 37.39 | < 0.001 | 37.26 | 0.001 | 82.19 | 0.001 | fiber |
| dose | 3.96 | < 0.001 | 4.14 | < 0.001 | 4.12 | 0.001 | | | dose |
| fiber:dose | 5.85 | < 0.001 | 5.99 | < 0.001 | 5.96 | 0.002 | | | fiber:dose |
| residuals | 55.75 | | 52.69 | | 52.67 | | 17.81 | | residuals |
| Lai et al., 2018 (data set 2) | | | | | | | Lai et al., 2018 (data set 2) | | |
| Model: OTU ~ diet*exercise | | | | | | | Model: OTU ~ diet_exercise | | |
| diet | 27.31 | < 0.001 | 30.50 | < 0.001 | 30.85 | 0.001 | 99.11 | 0.001 | diet |
| exercise | 12.18 | < 0.001 | 14.08 | < 0.001 | 13.99 | 0.001 | | | exercise |
| diet:exercise | 7.93 | < 0.001 | 7.51 | < 0.001 | 7.47 | 0.002 | | | diet:exercise |
| residuals | 52.31 | | 47.74 | | 47.70 | | 0.89 | | residuals |
| Le Sciellour et al., 2018 (data set 3) | | | | | | | Le Sciellour et al., 2018 (data set 3) | | |
| Model: OTU ~ diet*period + subject | | | | | | | Model: OTU ~ diet_period | | |
| diet | 1.78 | < 0.001 | 1.99 | < 0.001 | 2.14 | 0.001 | 13.33 | 0.001 | diet |
| period | 3.29 | < 0.001 | 3.45 | < 0.001 | 3.51 | 0.001 | | | period |
| diet:period | 1.32 | < 0.001 | 1.41 | 0.006 | 1.36 | 0.009 | | | diet:period |
| subject | 26.53 | < 0.001 | 26.42 | 0.073 | 27.28 | 0.046 | 86.67 | | subject |
| residuals | 66.13 | | 65.71 | | 65.71 | | | | residuals |
| Wang et al., 2016 (data set 4) | | | | | | | Wang et al., 2016 (data set 4) | | |
| Model: OTU ~ diet + time + diet:time + subject | | | | | | | Model: OTU ~ diet_time | | |
| diet | 2.49 | 0.067 | 2.11 | 0.038 | 2.27 | 0.036 | 4.32 | 0.143 | diet |
| time | 2.00 | 0.007 | 1.96 | 0.082 | 1.77 | 0.246 | | | time |
| diet:time | 5.15 | 0.824 | 4.41 | 0.777 | 4.45 | 0.701 | | | diet:time |
| subject | 50.44 | < 0.001 | 54.23 | < 0.001 | 64.30 | 0.001 | 95.68 | | subject |
| residuals | 31.23 | | 27.25 | | 27.21 | | | | residuals |
| Birkeland et al., 2020 (data set 5) | | | | | | | Birkeland et al., 2020 (data set 5) | | |
| Model: OTU ~ treatment:day + subject | | | | | | | Model: OTU ~ treatment_day | | |
| treatment: day | 1.75 | < 0.001 | 1.27 | 0.107 | 1.27 | 0.132 | -0.03 | 0.998 | treatment: day |
| subject | 69.38 | < 0.001 | 73.85 | 0 | 73.85 | 0.001 | | | subject |
| residuals | 28.88 | | 24.87 | | 24.87 | | | | residuals |

Distance-based ANOSIM and PERMANOVA and abundance-based ASCA and FFMANOVA were compared with respect to effect sizes (expressed as percentage of explained variance) and corresponding p-values.

[1]based on the 50–50 F-test, 999 permutations.

[2]based on 999 permutations.

[3]based on a combined factor with no interaction in the model (limitation of ANOSIM).

[clr]centred log-ratio transformed data as input.

**Data set 3**. In a longitudinal study by Le Sciellour *et al.* [49] the authors tested the effect of dietary fibre content on faecal microbiota in growing-finishing pigs fed alternately a low-fibre and a high-fibre diet during four successive 3-week periods. In this survey, the effect of diet was small (2%) compared to the effect of diet in data sets 1 and 2. Similarly, FFMANOVA and ASCA reported slightly lower p-values than PERMANOVA, but all three methods agreed on the effect of diet.

**Data set 4**. In a longitudinal study by Wang *et al.* [50] the objectives were to determine the impact of beta glucan on the composition of faecal microbiota in mildly hypercholesterolemic individuals. The individuals received for 5 weeks either a treatment containing 3 g high molecular weight (HMW), 3 g low molecular weight (LMW), 5 g LMW barley beta glucan or wheat and rice (control group) [50]. The effect of diet accounted for ~2% of the explained variance reported by FFMANOVA, ASCA and PERMANOVA. Diet was significant on a 5% level for PERMANOVA and ASCA (p = 0.036 and p = 0.038, respectively) and on a 10% level for FFMANOVA (p = 0.067). However, different conclusions were drawn with respect to time where significant result at the community level was obtained only for FFMANOVA (p = 0.007).

**Data set 5**. Birkeland *et al.* [44] assessed the effect of prebiotic fibres or a control supplement on faecal microbiota composition in human subjects with type two diabetes. The interaction effect of treatment and day accounted for 1–2% of the explained variance according to the FFMANOVA, ASCA and PERMANOVA. Significant results at the community level were obtained only for FFMANOVA (p < 0.001).

To summarise, PERMANOVA, FFMANOVA and ASCA gave almost identical results regarding effect sizes and statistical significance across studies (Table 2). They all revealed that there was a considerable difference in effect sizes of the main factor of interest (diet) between animal (2–37%) and human (1–2%) dietary interventions, with effect sizes being very small in human studies. In addition, three of the studies had crossover designs allowing for estimation of interindividual variation. This variation was considerably higher for trials involving human subjects (54–74%) compared to the animal study (26%). ANOSIM provided the most different results and was not able to reveal the same biological insights since the multifactorial nature of the studies cannot be taken into consideration by this approach.

## OTU level

Six of the methods, namely SIMPER, ASCA, FFMANOVA, ANCOM, ALDEx2 and DESeq2 can be used to make biological inferences for individual OTUs. The methods give different outputs which can be used to identify differentially abundant OTUs and/or rank the OTUs according to effect sizes (see Methods for details).

**Simulated data.**   The True Positive Rate (TPR) for the four simulated scenarios is shown in Fig 3. The True Negative Rate (TNR) was close to 100% for all methods and is therefore not shown. ASCA provided the overall best results in terms of the TPR in all four scenarios. FFMANOVA, SIMPER and ANCOM were all highly sensitive in the scenario with few significant OTUs and a high effect size (*"Few-High"*). FFMANOVA was also highly sensitive in the scenario with many significant OTUs and a high effect size (*"Many-High"*); SIMPER and ANCOM performed best in the scenario with few significant OTUs with a lower effect size (*"Few-Low"*). Both ALDEx2 and DESeq2 detected very few OTUs in any of the scenarios and therefore had very low TPR.

**Experimental data.**   Summary tables and scatterplots comparing ranking for different methods on the experimental data sets are given as S1 File and S1 Fig, respectively. The

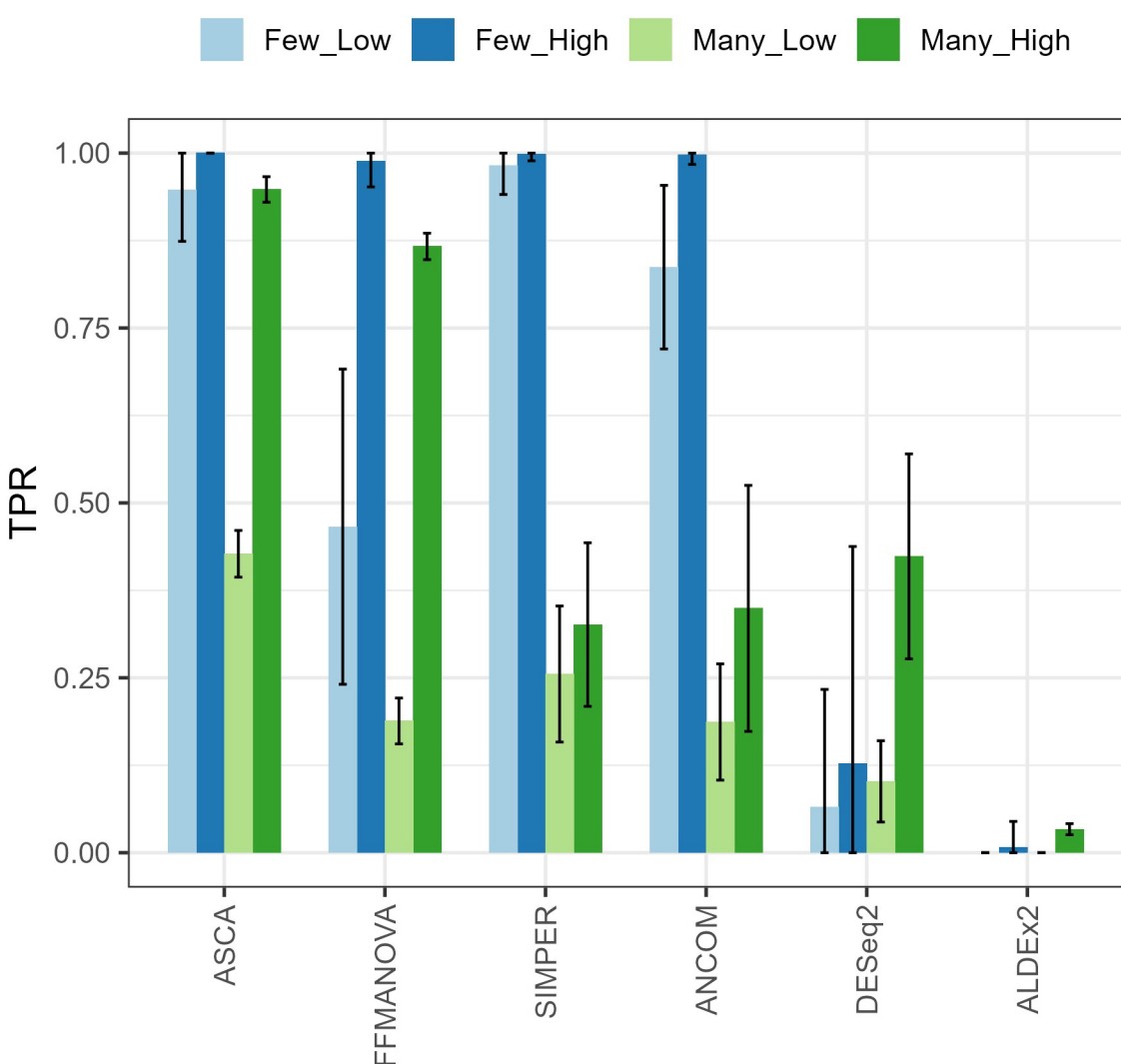

**Fig 3. Sensitivity (True Positive Rate) for the four scenarios in the simulation study.**

number of significant OTUs detected by the methods is summarised in S2 Table. As expected, many significant OTUs were discovered in the studies with large multivariate effect sizes (data sets Moen and Lai), whereas few OTUs were found in the studies with low multivariate effect sizes. The highest number of significant OTUs was identified by FFMANOVA, with almost twice the numbers detected by ALDEx2. ANCOM differed considerably between the study designs. ASCA recovered fewer OTUs than the other methods for the Moen data set, but more OTUs than the other methods for the other data sets.

**Correlation between the methods.** Agreement between the methods was investigated by calculating Spearman's rank correlation between all pairs of output metrics (Fig 4). In the simulated data, FFMANOVA and ASCA had higher agreement than any other pair of methods, with correlations ranging from 0.6 to 0.75. In addition, the correlations were generally higher for scenarios with many differentially abundant OTUs for all methods. The results from the experimental data sets also showed that FFMANOVA and ASCA had highest agreement, with correlations ranging from 0.4 to 0.9. However, correlations varied considerably between the

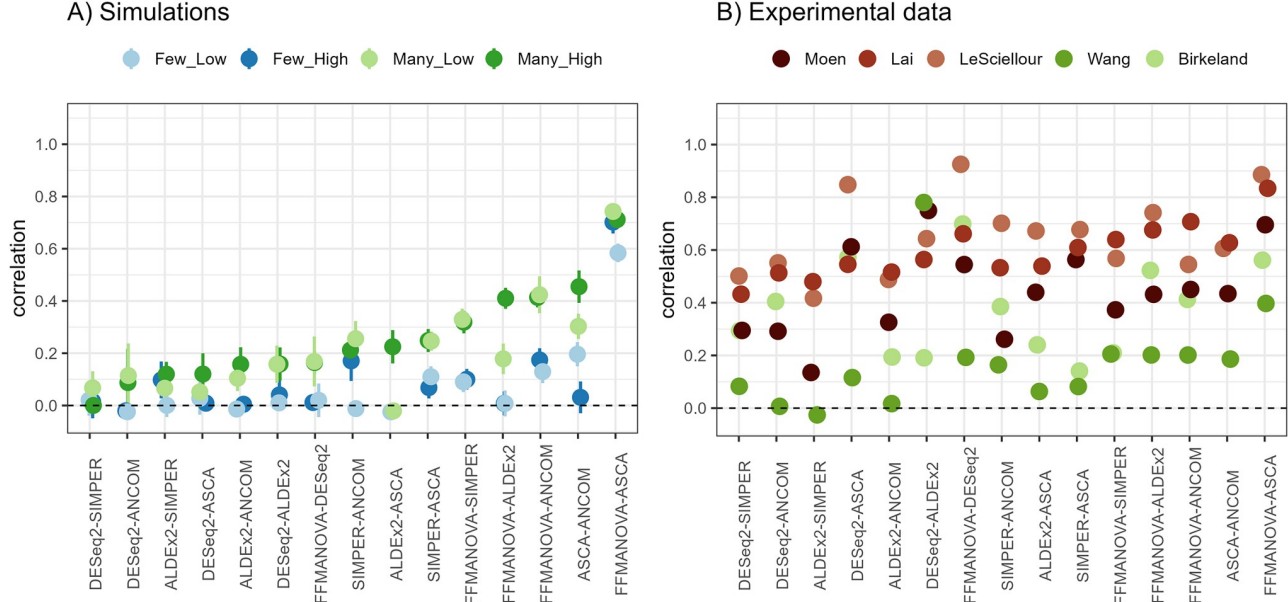

**Fig 4. Spearman's correlation (Y-axis) calculated for pairwise comparison of statistical methods (X-axis) for (A) simulated data and (B) five experimental data sets.** Each point represents Spearman's rank correlation coefficient between OTU ranking metrics from the two methods compared.

data sets. Highest agreements were observed for the animal studies, which are more controlled, and where interindividual variation is smaller compared to studies involving human subjects. The Moen and Wang data sets had a lower correlation between the methods than the other data sets for many comparisons, which could be explained by the fact that there are more than two levels of the diet factor, and omnibus tests will therefore not completely agree with the pairwise comparisons.

**Impact of OTU abundance.** Gut microbiome data are typically represented by a few dominant OTUs (relative abundance >1%) and a majority of low-abundant OTUs. One is therefore interested in ranking OTUs independent of their average abundance to detect biologically relevant changes in low-abundant OTUs. For all methods, except SIMPER, correlations in the range 0.1–0.4 were observed between the ranking statistics and the (log) mean relative abundance. However, it was not consistent between the data sets which method gave higher correlations. Correlations between the relative abundance and the ranking statistics were highest for the Moen data set and lowest for the Birkeland data set (S2 Table). ANCOM differed from the other methods as the ranking of OTUs was highly dependent on the abundance (Fig 5 and S1 Fig). In particular, the results indicated that highly abundant OTUs had either very high or very low W-stat, while low-abundant OTUs always had medium-to-low W-stat. To the best of our knowledge, this finding has not been reported before and shows that ANCOM is not able to identify changes in low-abundant OTUs. In the simulation study, clear differences between the scenarios with *"Few"* or *"Many"* differentially abundant OTUs can be observed (Fig 5A and 5C), and similar patterns are reflected for the experimental data (Fig 5B and 5D). It has been shown that power of ANCOM drops when the number of differentially abundant OTUs exceeds 25% [28]. In our *"Many"* simulations, 50% of the OTUs are differentially abundant, which justifies why ANCOM performed poorly for these scenarios. The dependency on the relative abundance can be more problematic also for the *"Many"* scenarios (Fig 5C) and data sets with many differentially abundant OTUs as, for instance, the Moen data set (Fig 5D).

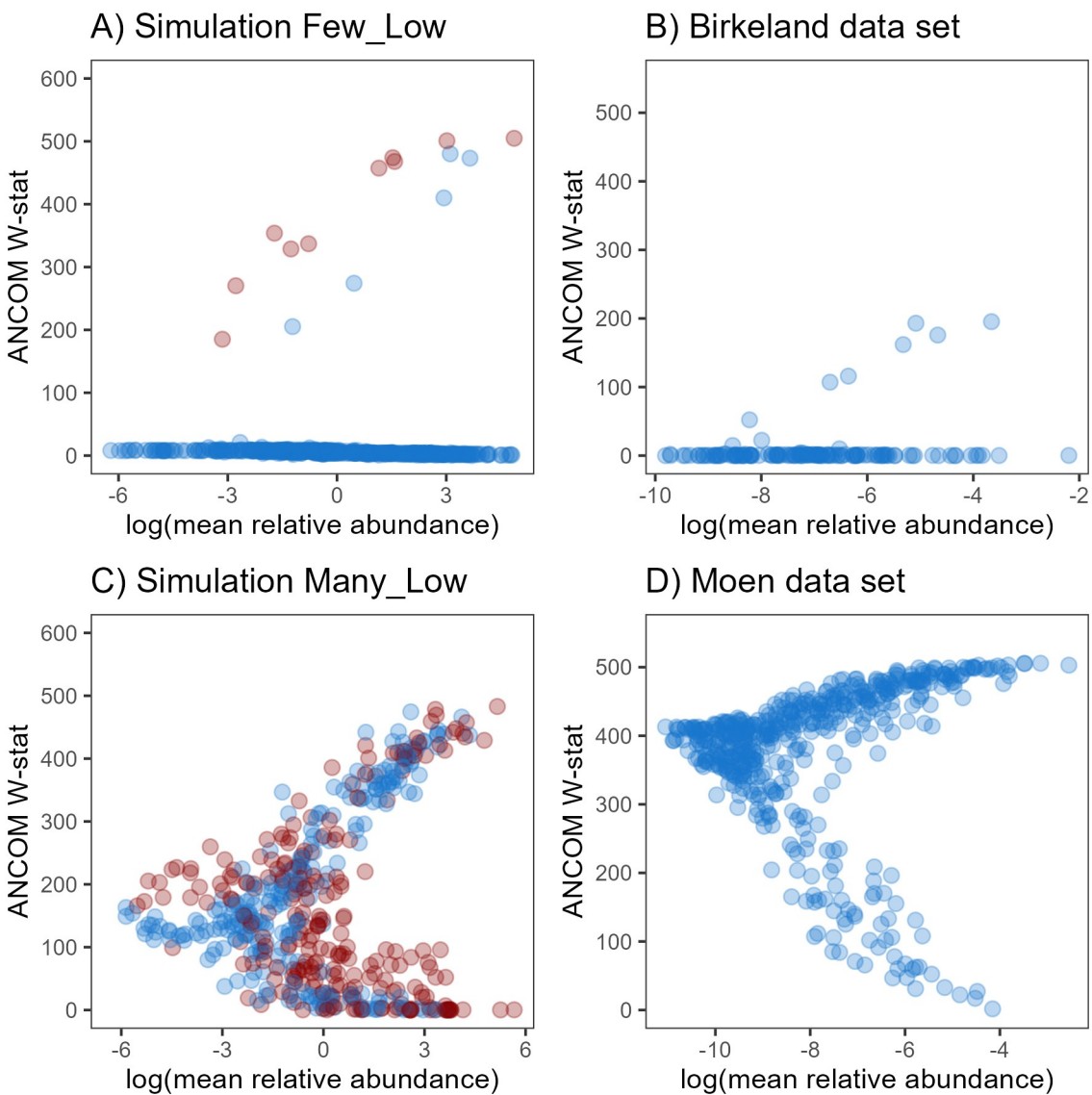

**Fig 5. Mean relative abundance (log-scale) plotted versus ANCOM W-stat.** (A) The *"Few-Low"* simulation scenario and (B) Birkeland data set, (C) *"Many-Low"* simulation scenario and (D) Moen data set. Red points (panels A and C) indicate differentially abundant OTUs.

## Discussion

Diet is considered to be an important driver of microbiome variation [51]. However, in observational population-based studies, diet consistently accounts for only a small proportion of microbiome variation, and this is partly due to large interindividual differences in microbiome composition, small sample sizes and limitations in study designs such as potentially insufficient washout periods in crossover studies [51–55]. In general, higher interindividual variation is observed in gut microbiome of human subjects compared to animal species [56]. This was confirmed by our results, and it can partly explain the lower diet related effect on the gut microbiomes in the two human studies. Variation in how much a diet can influence the microbiome is also dependent on the nutritional differences of the compared diets. Nevertheless, use

of animal models to study a causal role of the gut microbiome in health and disease is an established practice although animal models lack the specific interactions present in the complex system of a human organism [57].

Univariate and multivariate analyses provide information at different levels. Biologists will often find that the outputs of univariate analyses are easier to interpret compared to those generated by multivariate analyses, though assumptions are similar for both method types [10]. In general, multivariate methods provide a more holistic overview of differences between samples and account for correlations and interactions between the variables, whereas univariate methods are well suited to point out the differences for specific microbial groups. Therefore, the two levels of analysis provide complementary information, and it is generally of biological interest to report differences at both levels.

FFMANOVA and ASCA consider the covariance between all OTUs. It could therefore be expected that these methods would be better at detecting scenarios with many differentially abundant OTUs due to the *consistency at large* phenomenon, i.e. that many OTUs carry the same information and such methods collectively are able to detect small effects that are not significant at the univariate level [58, 59]. At the community level, these methods performed similarly to the distance-based methods. At the OTU level, they had considerably higher sensitivity to identify true positive OTUs than the other methods in the *"Many"* OTU scenarios.

FFMANOVA and ASCA depend on the relative scaling of the OTUs, while for the distance-based methods it depends on the chosen distance measure. It is a common practice in many areas to scale all variables to equal variance thus giving them an equal weight in the model, but other options are also possible, see, for instance, van den Berg *et al.* [60]. The clr-transformation puts the variables at comparable levels, and the need for scaling is less obvious. However, the highly abundant OTUs might still have slightly higher variance and scaling should be considered depending on the data characteristics and the biological interpretation. With scaling, all OTUs have the same contribution in the analysis, whereas without scaling the more abundant OTUs will dominate the analyses and the inferences will be related to the more abundant OTUs.

It is a known fact that microbial sequence data are zero-inflated, and rare OTUs should be removed prior to downstream statistical analyses. We have observed that the threshold for filtering out OTUs can significantly affect the results both at the community and OTU levels. This can be exemplified in the human Birkeland data set, where stricter OTU-filtering performed in the present study resulted in no significant treatment effect by ASCA, in contrast to the original publication [44].

The tools tested in the present study vary in flexibility. SIMPER allows only pairwise comparisons and ANOSIM provides multigroup analysis, but it is not suited for multifactorial study designs. The other methods can employ more complex models with multiple factors with varying number of levels, and corresponding interactions. In experiments with repeated measurements, the subject effect is often included as a random factor. Neither of the methods discussed here can do this, hence the subject effect was included as a fixed factor. These aspects should be considered when selecting methods because different study designs might require different types of statistical models and tests. Newer developments of ASCA, namely ASCA+ [61] and LiMM-PCA [62], increase flexibility when there are unbalanced designs or random (i.e. subject) effects, respectively. Even so, the longitudinal modelling of large microbiome data sets in combination with multiple covariates is only starting to emerge [6].

Although Spearman's rank correlation indicated good agreement for the animal studies (Fig 4), little overlap between lists of "significant" results could be detected (results not shown). There can be several reasons for this. One important aspect is that different criteria must be used to define "significance" or generally "importance". FFMANOVA, SIMPER,

DESeq2 and ALDEx2 provide p-values, while more heuristic tools must be applied with ASCA and ANCOM. For some methods, such as FFMANOVA and ANCOM, the ranking is related to *all* levels of the experimental factors, whereas the other methods use only pairwise comparisons (SIMPER) or comparison against a selected control/reference level (ALDEx2 and DESeq2). ASCA provides a test related to all levels at the community level, while the PLS-DA step relates to pairwise comparisons at the OTU level. This will introduce a bias for data sets with multilevel factors where more than one level is different from each other. Nevertheless, we chose to compare these rankings as this is the output that is available to the user. In the simulations only one level of one factor was designed to have an effect, hence the "omnibus" and the "specific" tests are more directly comparable. In experiments with multilevel factors, additional information can be obtained by looking at the clr-difference between the groups of interest in addition to the ranking statistics.

Moreover, differences in sample collection, sample preparation and sequencing contribute to additional variability, which, in turn, affects the validity of the results [63] and complicates comparisons across studies with similar interventions.

Past benchmarking studies [9, 64, 65] have reported varying results from different tools, which was also confirmed in our study. Currently, there is no consensus for the best existing tool for detecting differentially abundant microbial taxa, and there is no reason to believe that one single method is best in all cases. Based on our simulations, the generic multivariate tools, ASCA and FFMANOVA, performed best. We anticipate that these results will inform future studies with more complex settings where more than one factor has an effect or interactions between experimental factors are included in the model.

In addition to performance, ease of use is an important aspect when selecting the appropriate tool. FFMANOVA and ASCA are based on standard statistical tools, namely PCA and ANOVA. Some of the tools designed for microbiome high-throughput sequencing data, on the other hand, can be difficult for non-statisticians, and it can be questioned if the users are able to interpret all parameters correctly, even if the methods are supported by comprehensive documentation. It is always good scientific practice to compare and report outputs from several methods. There are no standards on how to report multiple modelling results, and there is a high risk of "fishing for significance" when several methods are applied [66]. Before designing the experiment, researchers should be aware of the different properties of the statistical methods and consider whether it is most important not to miss out on any possible findings or to obtain robust results. In the latter case, OTUs should be reported as differentially abundant only if they were flagged as "significant" by several methods [66].

## Conclusion

In the present study, we compared the performance of several multivariate ANOVA-like statistical methods taking four simulated scenarios and five real dietary intervention microbiome data sets as examples. At the community level, all the different methods came to similar conclusions; at the OTU level, the agreement between the methods considerably varied. ANCOM provided output metrics that were dependent on the average abundance, making it impossible to detect differences in low-abundant OTUs. At the OTU level, the ranking of OTUs obtained with different methods correlated better for animal studies than for human studies, possibly due to lower interindividual variation in animal studies. Based on the simulation results we advise applying FFMANOVA and ASCA for overall and pairwise comparisons of microbiome data, respectively, also because these methods provide output at both the community and OTU levels, can handle several design factors, as well as other data types common in microbiome research.

## Methods

### Experimental design and data characteristics

The dietary intervention data sets were selected based on the study design, with a minimum of two independent variables (for example, diet and dose; see S1 Table for details). Prior to the statistical analyses, the data were filtered to keep the OTUs that were present: (1) with relative abundance more than 0.005% in an individual, and (2) in at least 50% of the individuals and in one of the groups. For each study in total 507, 561, 560, 397 and 216 OTUs passed this filter and were subsequently used in downstream analyses (S1 Table and S2 File). All statistical analyses were performed in R version 4.1.0 [67] unless otherwise specified and were run in 999 permutations.

**Simulated data** were generated using the R package metaSPARSim [68] developed for simulating 16S rDNA data. The simulation was a two-step process: (1) modelling the abundance (expected counts for each experimental group) using a Gamma distribution; (2) modelling the within-group variability using a Multivariate Hypergeometric (MHG) distribution taking the output from step 1 as input parameters for the distribution. In addition, metaSPARSim contains functions for parameter estimation from observed data. We used data set 1 [42] (S1 Table) to estimate real starting parameters for the simulations. Raw counts were generated for the same number of OTUs as in the experimental data set, and subsequently pre-processed and analysed in the same manner as for the experimental data (described below). Four different scenarios were simulated, with a varying number of the differentially abundant OTUs (*"Few"* versus *"Many"*) and the effect sizes (*"Low"* versus *"High"*). 100 data sets were generated and analysed for each of the four scenarios:

1. *"Few—Low"*: 10 randomly selected OTUs were assigned random log2 fold changes from a uniform distribution with boundaries [3,4].

2. *"Few—High"*: 10 randomly selected OTUs were assigned random log2 fold changes from a uniform distribution with boundaries [8,9].

3. *"Many—Low"*: 254 randomly selected OTUs were assigned random log2 fold changes from a uniform distribution with boundaries [3,4].

4. *"Many—High"*: 254 randomly selected OTUs were assigned random log2 fold changes from a uniform distribution with boundaries [8,9].

**Zero-value replacements** were done prior to clr-transformation [27, 69] by applying function *cMultRepl* with a setting *method = 'CZM'* in the R package zCompositions [70]. Zero replacement is an ongoing and yet unsolved statistical problem in microbiome research, and newer methods are constantly being developed and applied to both simulated and experimental data sets [71–74].

**ANOSIM** and **PERMANOVA** were run using the functions *anosim* and *adonis* in the R package vegan [75], with a setting *method = Euclidean* and the clr-transformed data as input.

**SIMPER** was run on filtered relative abundance data using function *simper* from the R package vegan [75]. The p-values for each OTU between selected pairwise comparisons of diet levels were used as a ranking metric; the p-value represents the probability of getting a larger contribution to the Bray-Curtis dissimilarity in a random permutation of the group factor.

**FFMANOVA** was performed by using the function *ffmanova* implemented in the R package ffmanova [35] on the clr-transformed and standardised data. Raw p-values were used as a ranking metric.

**ASCA** was run on the clr-transformed and centred data using an in-house implementation in MATLAB (R2018b, The MathWorks Inc.); p-values at the community level were calculated

by permutation tests ($n$ = 999). PLS-DA models [76] with the ASCA diet effect matrix, residuals as a predictor and factor levels as a response were used for pairwise comparison of diet levels. Variable Importance in Prediction (VIP) values [76] were used to identify significant OTUs, and the VIP threshold was set using the Uninformative Variable Elimination (UVE) method [77]. The UVE procedure was repeated 100 times, and OTUs that were above the threshold in >95% of the repetitions were defined as significant. For study designs with two diet levels, the ASCA loadings were used to rank OTUs, while PLS-DA regression coefficients were used for pairwise comparisons of multilevel diet factors.

**ANCOM** was run by using the ANCOM 2.0 source code implemented in R [28] using filtered relative abundance data as input. The W-stat was used to rank OTUs indicating the number of significantly different pairwise log-ratios while adjusting for FDR by applying a Benjamini-Hochberg correction at a 0.05 level of significance.

**ALDEx2** was run using the functions *aldex.clr* and *aldex.glm* from the R package ALDEx2 v.1.18.0 [26]. We used raw counts as input and p-values for selected factor level contrasts to rank OTUs.

**DESeq2** was run using the functions *DESeqDataSetFromMatrix* (to generate object), *DESeq* (for analysis) and *results* (to extract results) from the Bioconductor package version 1.32.0 [30]. We used raw counts as input and raw p-values for selected level contrasts to rank OTUs (default settings).

## Factor level comparisons

For study designs with more than two diet groups, only one pairwise comparison was analysed for the methods based on the pairwise group comparison, namely ALDEx2, ASCA and SIMPER. The following pairs with the most contrasting outcomes were compared: (1) BSG group vs. IN group [42]; (2) control group vs. 3 g HMW [50] and (3) Placebo 6 weeks group vs. Fibre 6 weeks group [44].

**Differentially abundant OTUs** were identified by setting thresholds on the ranking metrics for each method. For the methods providing p-values, the threshold was set to 0.01. For ANCOM, the 60th percentile of the empirical distribution of the W-stat was used as a threshold. For ASCA, OTUs selected in 95 out of 100 UVE-runs were identified as differentially abundant.

**True Positive Rate (TPR) and True Negative Rate (TNR)** were calculated for the simulated data. TPR (also called *Power* or *Sensitivity*) is calculated as *TP/P*, where *TP* is the number of true differentially abundant OTUs identified by a statistical method and *P* is the number of differentially abundant OTUs defined in the simulation setup. The TNR (also called *Specificity*) is calculated as *TN/N*, where *TN* is the number of true non-differentially abundant OTUs identified by a statistical method and *N* is the corresponding number defined in the simulation setup.

## Supporting information

**S1 Fig. Pairwise scatterplots between OTU ranking metrics for each of the statistical methods.** The (A) Moen, (B) Lai, (C) Le Sciellour, (D) Wang and (E) Birkeland data sets. (PDF)

**S2 Fig. Explained variance for simulated data and the number of simulations where the simulated effect was detected.** Differentially abundant OTUs are shown in red and non-differentially abundant OTUs in black. (PDF)

**S1 Table. An overview of the experimental studies and their data characteristics.** (XLSX)

**S2 Table. Number and per cent of significant OTUs detected by each method.**
(XLSX)

**S1 File. Summary table of the results obtained for the OTU level across five experimental data sets for the methods SIMPER, ASCA, FFMANOVA, ANCOM, ALDEx2 and DESeq2.**
(XLSX)

**S2 File. Filtered relative abundance data and metadata for five experimental data sets.**
(XLSX)

**S3 File. RData file with raw counts, relative abundance and clr-transformed and filtered data for the Moen data set.**
(RDATA)

**S4 File. R script for the Moen data set analyses.**
(R)

**S5 File. R script for the simulated data analyses.**
(R)

**S6 File.**
(DS_STORE)

**S7 File.**
(RHISTORY)

## Acknowledgments

We thank all the authors who were involved in data generation used in the present study. All data sets are properly cited and referred to in S1 Table.

## Author Contributions

**Conceptualization:** Ingrid Måge, Ida Rud, Ingunn Berget.

**Data curation:** Maryia Khomich.

**Formal analysis:** Maryia Khomich, Ingrid Måge, Ingunn Berget.

**Funding acquisition:** Ingrid Måge, Ida Rud.

**Investigation:** Maryia Khomich, Ingrid Måge, Ida Rud, Ingunn Berget.

**Methodology:** Maryia Khomich, Ingrid Måge, Ida Rud, Ingunn Berget.

**Project administration:** Maryia Khomich, Ingrid Måge, Ida Rud.

**Resources:** Ida Rud.

**Software:** Maryia Khomich, Ingrid Måge, Ingunn Berget.

**Supervision:** Ingrid Måge, Ida Rud, Ingunn Berget.

**Validation:** Maryia Khomich, Ingrid Måge, Ingunn Berget.

**Visualization:** Maryia Khomich, Ingrid Måge, Ida Rud, Ingunn Berget.

**Writing – original draft:** Maryia Khomich, Ingrid Måge, Ida Rud, Ingunn Berget.

**Writing – review & editing:** Maryia Khomich, Ingrid Måge, Ida Rud, Ingunn Berget.

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
