## [Decision Letter · Decision Letter 0]

19 Apr 2021

PONE-D-21-07978

Analysing microbiome intervention design studies: Comparison of alternative multivariate statistical methods

PLOS ONE

Dear Dr. Khomich,

Thank you for submitting your manuscript to PLOS ONE. After careful consideration, we feel that it has merit but does not fully meet PLOS ONE’s publication criteria as it currently stands. Therefore, we invite you to submit a revised version of the manuscript that addresses the points raised during the review process.

We look forward to receiving your revised manuscript.

Kind regards,

Lingling An

Academic Editor

PLOS ONE

Journal Requirements:

Please include captions for your Supporting Information files *at the end of your manuscript*, and update any in-text citations to match accordingly. Please see our Supporting Information guidelines for more information: http://journals.plos.org/plosone/s/supporting-information.

Reviewers' comments:

Reviewer's Responses to Questions

**Comments to the Author**

1. Is the manuscript technically sound, and do the data support the conclusions?

Reviewer #1: Yes

Reviewer #2: Partly

Reviewer #3: Yes

2. Has the statistical analysis been performed appropriately and rigorously? 

Reviewer #1: Yes

Reviewer #2: No

Reviewer #3: Yes

3. Have the authors made all data underlying the findings in their manuscript fully available?

Reviewer #1: Yes

Reviewer #2: Yes

Reviewer #3: Yes

4. Is the manuscript presented in an intelligible fashion and written in standard English?

Reviewer #1: Yes

Reviewer #2: Yes

Reviewer #3: Yes

5. Review Comments to the Author

Reviewer #1: In this article, the authors conducted a detailed comparison between different statistical methods for microbiome data analysis. The comparison is based on some dietary intervention studies. I enjoy reading this paper and think such a comparison is needed. But there is still a large room to improve in numerical comparison and writing. See details in attachment.

Reviewer #2: In Khomich et al., the authors compared a variety of distance-based and OTU-level microbiome testing methods, on five human- and animal-based dietary studies. The authors found that, on these datasets, the distance-based methods largely agree regarding both effect size and statistical significance, whereas large discrepancies exist for OTU-level testing methods.

I found the authors investigation here helpful, especially with the focus on dietary intervention studies. The empirical evaluation of the methods’ performance will provide helpful guidance for researchers in the field. However, I do find the evaluation efforts limited, in order to make rigorous claims comparing across methods. I think additional simulation and real dataset based analysis will significantly boost the paper’s usefulness to interested readers. My comments are detailed below.

Major

1. Most methods evaluation efforts, including those published for microbiome data, include simulation sections. The authors correctly pointed out that simulation evaluation has its own caveats, such as the data simulation model favoring some evaluated methods over others. However, one of the most important benefits of simulated datasets is that the ground truth (associated OTUs, effect sizes, etc.) are known, which is impossible for real-world datasets. This importantly enables rigorous evaluation of statistical performance, such as power and false positive rates, as consistent metrics across methods. In order to avoid favoring one of the evaluated methods by using a similar simulation model, nonparametric methods such as shuffling the data or combining different environments can be considered. See e.g. PMID 24699258, or the Thorsen et al. paper the authors cited in the work.

I would strongly advise the addition of such efforts, as they can provide valuable insights into actual performance of between methods, as the current claims are limited to e.g. “A and B methods are more similar than C method”. The latter is still valuable information, but it does not necessarily inform the reader as to if A/B are preferable to C.

2. Relatedly, there is space for improvement in the real-world data based evaluations. I think one of the strong suits of the authors’ efforts is the inclusion of a variety of dietary intervention studies, with different study designs and host environments. However, the authors claims are limited to which methods performed more similarly in which datasets, using also limited evaluation metrics (p-value and effect size for dissimilarity based analysis, feature rank Spearman correlation for per-OTU analysis). Are the reported percentage variability explained largely consistent with previous literature? Are the important features identified by each method actually biologically plausible? Answering these questions can provide important insight on the validity of each method.

Moderate:

1. I’m not sure if the comparison for Figure 2 was carried out entirely correctly. That is, I wonder if the authors might have rankings that are not comparable between studies. E.g. line 304-306: “Also note that for some methods (FFMANOVA, ALDEx2 and ANCOM), the ranking is related to all levels of the experimental factors, whereas the other methods use pairwise comparisons only.” Pairwise comparison and all-level comparison are essentially different tests, and rankings using their test statistics are not directly comparable. The authors should be careful to compare the same tests across methods.

2. The authors claimed LEfSe can only be compared against other methods in two datasets, as the other datasets report too few significant results. However, results for all features can be reported by LEfSe, by setting the most lenient effect size and p-values thresholds. Focusing on datasets with only significant features can artificially inflate LEfSe’s agreement with other methods, as significant features should be definition have more confidence in their effect sizes.

Minor:

1. I wonder if the visualization for Figure 2 can be improved. The color can be arranged for animal-based and human-based studies to be more similar among each other. Alternatively, the authors might instead consider plotting five method-against-method heat maps, one for each study, to make the current Y-axis less dense.

Reviewer #3: Khomich et. al. provided a review and method comparisons of existing microbiome analysis tools for analyzing intervention design studies. The paper is well-written. They benchmarked four datasets. The analysis was well-conducted. I have some suggestions below:

1. Not very clear to me why intervention design studies were different from there microbiome studies with categorical covariates.

2. I would appreciate the authors to review the statistical assumptions that are adopted in each of these methods. The review of the methods is very descriptive. For example:

“Among abundance-based beta diversity indices, Bray-Curtis is a common choice due to its theoretical properties and empirical accuracy. Widely applied phylogenetic beta diversity indices is UniFrac-type metrics. However, UniFrac is unsuitable as a distance metric for studies with 84 small sample size, which is usually the case for dietary intervention trials21,22.”

What does theoretical properties mean? Statistical properties? What does “Empirical accuracy” refer to?

Since this is a rather method paper, author should at least explain the method framework and how the advantages and disadvantages related to different methods.

3. There are many places that authors commented on correlation between taxa, but they did not discuss about in their four real datasets, how correlated the taxa are? Did they on purposely choose the ones contain different level of correlation to distinguish results?

4. Bioinformatic pipeline and data quality that used to pre-processing can make differences for the downstream analysis for microbiome studies. Any suggestions on statistical tools which can be better/worse in these situations?

6. PLOS authors have the option to publish the peer review history of their article (what does this mean?). If published, this will include your full peer review and any attached files.

Reviewer #1: No

Reviewer #2: No

Reviewer #3: No

---

## [Author Response · Author response to Decision Letter 0]

14 Sep 2021

INTRO

Reviewer #1: In this article, the authors conducted a detailed comparison between different statistical methods for microbiome data analysis. The comparison is based on some dietary intervention studies. I enjoy reading this paper and think such a comparison is needed. But there is still a large room to improve in numerical comparison and writing. See details in attachment. The comments from the attached .pdf-file are copied and answered below.

Thank you for your interest in our work, comprehensive and constructive comments that have helped us to improve the current version of the manuscript. A detailed response to all comments is provided below.

Several methods suggesting the same conclusion doesn’t imply that they are reliable. All methods may be wrong in the same way! It might be better to benchmark the analysis with some special experiment design or simulation study on real data.

We have conducted a simulation study imitating four different scenarios to investigate how the tested statistical methods perform in situations with varying effect sizes and different numbers of differentially abundant OTUs. The simulated data were based on data set 1 (Moen et al., 2016) using the same study design and OTU counts as a starting point. At the community level, the methods strongly agreed and there was very low variability between simulations on the estimated explained variance. At the OTU level, the ASCA and FFMANOVA had the best performance in terms of True Positives and False Negatives.

We have added the description and outcomes of the simulations throughout the revised manuscript. We have also made new figures visualising the outcomes of the simulations.

In the OTU level comparison, several other popular differential abundance tests should be also included into the comparison. For example, DESeq2 (Love et al., 2014), MetagenomeSeq (Paulson et al., 2013), ANCOM-BC (Lin and Peddada, 2020), DR (Morton et al., 2019).

We have carefully checked the available literature on the differential abundance (DA) methods and selected a subset for our study. The main novelty of our work lies in comparing the less known and more general methods FFMANOVA and ASCA to more established methods designed for microbiome data. Many of the available methods, such as ANCOM (Mandal et al., 2015), DESeq2 (Love et al., 2014), edgeR (Robinson et al., 2010) and MetagenomeSeq (Paulson et al., 2013) have extensively been compared (Weiss et al., 2017; Lee et al., 2017). Both studies highlighted ANCOM as the best method regarding sensitivity and false discovery rate (FDR). Therefore, in our study we included ANCOM as the best performing method. But our first submission was done before ANCOM-BC (Lin and Peddada, 2020) was published. The paper shows that ANCOM and ANCOM-BC have comparable FDR and power, and we therefore think that our results on ANCOM are representative also for ANCOM-BC.

Also, the scope of our paper is limited to methods suited for analysing multifactor experimental designs. MetagenomeSeq (Paulson et al., 2013) and DR (Morton et al., 2019) are developed for one-factor experiments. Because of this, we have also chosen to exclude LEfSe from our revised manuscript. Based on the study by Love et al. (2014), DESeq2 and edgeR use a similar modelling strategy (negative binomial) and perform similarly in simulations, but differ in normalisation, outlier handling, and other adjustable parameters. Thus, we decided to include only one of the methods – DESeq2 – because differences between DESeq2 and edgeR are at a different conceptual level rather than the differences between, for instance, ASCA and these methods.

In distance-based methods, the choice of distance is very important. It would be great to compare different choices of distance.

We know that the choice of distance is very important in microbiome surveys. However, this aspect of analysis was beyond the scope of this study. We chose Bray-Curtis distance because it is the most used beta-diversity metric, and the first author of the manuscript has previously published a study where the performance of four beta-diversity metrics, i.e., Bray-Curtis, Jaccard, Gower and Raup-Crick (as implemented by the “bray”, “jaccard”, “gower” and “raup” options for the vegdist function in R package vegan), was compared on a set of microbiome data in eight different variations using non-metric multidimensional scaling (NMDS) ordination with two dimensions (Supplementary Figure S4; Khomich et al., 2017: https://doi.org/10.1016/j.funeco.2017.01.008). Assessment of metrics’ validity was done by Procrustes correlation tests run in 999 permutations (function procrustes in R package vegan). Jaccard, Raup-Crick and Bray-Curtis dissimilarity indices produced very similar results, with Gower being the least robust metric.

In abundance-based methods, phylogenetic tree information can also be incorporated (Morton et al., 2017; Wang et al., 2020). They should be cited.

Thank you for your suggestion. We cited these papers in our revised manuscript (see lines 67-68, second paragraph of Introduction).

In most of abundance-based methods, the zeros are usually replaced by some small positive constant. This is a very important problem in microbiome data analysis. Currently, the authors only consider one imputation method. It would be better to include a separate experiment to compare different ways to handle zeros. Some studies (Costea et al., 2014) suggests that handling zeros can affect the analysis a lot. Some recent statistical methods (Brill et al., 2019; Wang et al., 2021) are designed for zero problem should be at least cited.

Zero imputation is an ongoing and yet unsolved statistical problem in microbiome research, and newer statistical methods are constantly being developed and applied to both simulated and experimental data sets. We agree with the Reviewer that zero imputation can impact the results and therefore should be considered, but it was beyond the scope of this study. We cited the suggested literature in our revised version of the manuscript (see lines 470-473).

Reviewer #2: In Khomich et al., the authors compared a variety of distance-based and OTU-level microbiome testing methods, on five human- and animal-based dietary studies. The authors found that, on these datasets, the distance-based methods largely agree regarding both effect size and statistical significance, whereas large discrepancies exist for OTU-level testing methods.

I found the authors investigation here helpful, especially with the focus on dietary intervention studies. The empirical evaluation of the methods’ performance will provide helpful guidance for researchers in the field. However, I do find the evaluation efforts limited, to make rigorous claims comparing across methods. I think additional simulation and real dataset-based analysis will significantly boost the paper’s usefulness to interested readers.

We agree with the Reviewer and have conducted simulation studies imitating four different scenarios. The simulated data were based on data set 1 (Moen et al., 2016) using the same study design and OTU counts as a starting point. At the community level, the methods strongly agree and there is very low variability between simulations on the estimated explained variance. At the OTU level, the ASCA and FFMANOVA had the best performance in terms of True Positives and False Negatives. We added the description and outcomes of the simulations throughout the revised version of the manuscript. We have also made new figures depicting the simulation results.

MAJOR

1. Most methods evaluation efforts, including those published for microbiome data, include simulation sections. The authors correctly pointed out that simulation evaluation has its own caveats, such as the data simulation model favoring some evaluated methods over others. However, one of the most important benefits of simulated datasets is that the ground truth (associated OTUs, effect sizes, etc.) are known, which is impossible for real-world datasets. This importantly enables rigorous evaluation of statistical performance, such as power and false positive rates, as consistent metrics across methods. To avoid favoring one of the evaluated methods by using a similar simulation model, nonparametric methods such as shuffling the data or combining different environments can be considered. See e.g., PMID 24699258, or the Thorsen et al. paper the authors cited in the work. I would strongly advise the addition of such efforts, as they can provide valuable insights into actual performance of between methods, as the current claims are limited to e.g. “A and B methods are more similar than C method”. The latter is still valuable information, but it does not necessarily inform the reader as to if A/B are preferable to C.

We agree with the Reviewer and have conducted simulation studies imitating four different scenarios. The simulated data were based on data set 1 (Moen et al., 2016) using the same study design and OTU counts as a starting point. At the community level, the methods strongly agree and there is very low variability between simulations on the estimated explained variance. At the OTU level, the ASCA and FFMANOVA had the best performance in terms of True Positives and False Negatives. We added the description and outcomes of the simulations throughout the revised version of the manuscript. We have also made new figures depicting the simulation results.

2. Relatedly, there is space for improvement in the real-world data-based evaluations. I think one of the strong suits of the authors’ efforts is the inclusion of a variety of dietary intervention studies, with different study designs and host environments. However, the authors claims are limited to which methods performed more similarly in which datasets, using also limited evaluation metrics (p-value and effect size for dissimilarity-based analysis, feature rank Spearman correlation for per-OTU analysis). Is the reported percentage variability explained largely consistent with previous literature? Are the important features identified by each method actually biologically plausible? Answering these questions can provide important insight on the validity of each method.

In our simulation study, the reported explained variance was very consistent across 100 simulations and strongly agreed between the tested methods. Please see the following examples in the literature: (1) Vandeputte et al. (2017): http://dx.doi.org/10.1136/gutjnl-2016-313271 and (2) Måge et al. (2018): https://www.biorxiv.org/content/10.1101/363630v1.

MODERATE

1. I’m not sure if the comparison for Figure 2 was carried out entirely correctly. That is, I wonder if the authors might have rankings that are not comparable between studies. E.g., line 304-306: “Also note that for some methods (FFMANOVA, ALDEx2 and ANCOM), the ranking is related to all levels of the experimental factors, whereas the other methods use pairwise comparisons only.” Pairwise comparison and all-level comparison are essentially different tests, and rankings using their test statistics are not directly comparable. The authors should be careful to compare the same tests across methods.

We agree that pairwise and all-level comparisons are not directly comparable. If one chooses to use a method that produces all-level comparisons, one will use them to interpret the results, and vice versa, if one uses a method that only outputs pairwise comparisons, they will be used in the interpretation of the results. Our simulation study was designed with only one factor level different from all others. In this case, the all-level and pairwise comparisons are comparable.

We rewrote the paragraph in the revised manuscript (see lines 166-178), and now it is read as follows:

“The aim of the method comparison was to investigate how different strategies for statistical modelling affect biological inference. At the community level, methods were compared with respect to effect sizes (expressed as percentage of explained variance) and corresponding p-values. At the OTU level, comparison of methods is complex because some methods provide results for an omnibus test of differences between factor levels (FFMANOVA and ANCOM), whereas the other methods provide ranking for specific pairwise comparisons (ASCA and PLS-DA, SIMPER) or contrasts/model coefficients (ALDEx2 and DESeq2). Even so, a biologist will make inferences based on the output provided by the chosen method, and in this context, it is relevant to compare the ranking statistics although the tests are not the same. In our study, the ranking of OTUs was compared by Spearman’s rank correlation and by investigation of scatterplots between the different ranking metrics. For the simulated data, where we know which OTUs are differentially abundant, True Positive Rate (TPR) and True Negative Rate (TNR) were also evaluated.”

2. The authors claimed LEfSe can only be compared against other methods in two datasets, as the other datasets report too few significant results. However, results for all features can be reported by LEfSe, by setting the most lenient effect size and p-values thresholds. Focusing on datasets with only significant features can artificially inflate LEfSe’s agreement with other methods, as significant features should by definition have more confidence in their effect sizes.

LEfSe is a stepwise approach that combines univariate analysis with multivariate discriminant analysis. Even though the method has found wide application in microbiome research, it is not adapted to experimental designs with several multilevel factors and is therefore omitted from the revised version of the manuscript.

MINOR

1. I wonder if the visualization for Figure 2 can be improved. The color can be arranged for animal-based and human-based studies to be more similar among each other. Alternatively, the authors might instead consider plotting five method-against-method heat maps, one for each study, to make the current Y-axis less dense.

Figure 4 (in the revised manuscript) is a two-panel figure: (A) simulated data and (B) experimental data. We implemented the Reviewer´s suggestion: the colour is currently arranged for animal- and human-based studies to be more similar among each other (brown vs. green, respectively; see Figure 4B). The method pairs (X-axis) are sorted from low to high correlation between the methods (Y-axis). We have also plotted this figure as a heatmap, but the visualization was less user-friendly, therefore we decided to keep the original dotted plot with the suggested modifications by the Reviewer.

Reviewer #3: Khomich et. al. provided a review and method comparisons of existing microbiome analysis tools for analyzing intervention design studies. The paper is well-written. They benchmarked five datasets. The analysis was well-conducted. I have some suggestions below:

1. Not very clear to me why intervention design studies were different from the microbiome studies with categorical covariates.

In contrast to observational studies, dietary intervention trials are usually small in sample size but performed in (semi)-controlled environments and tailored to a specific research hypothesis. The studies often include multiple experimental factors, possibly with more than two levels. This information can be found in the revised manuscript (lines 69-72).

2. I would appreciate the authors to review the statistical assumptions that are adopted in each of these methods. The review of the methods is very descriptive. For example: “Among abundance-based beta diversity indices, Bray-Curtis is a common choice due to its theoretical properties and empirical accuracy. Widely applied phylogenetic beta diversity indices is UniFrac-type metrics. However, UniFrac is unsuitable as a distance metric for studies with small sample size, which is usually the case for dietary intervention trials.” What does theoretical properties mean? Statistical properties? What does “Empirical accuracy” refer to? Since this is a rather method paper, author should at least explain the method framework and how the advantages and disadvantages related to different methods.

Initially we did consider including more details about each method but decided not to include it to keep the manuscript concise. We chose instead to refer to the original publications for further details about statistical assumptions. The sentence about the beta diversity metrics was rephrased and can be read as follows (lines 87-91):

“Among abundance-based beta diversity indices, Bray-Curtis is the most common choice for count data. The most widely applied phylogenetic beta diversity indices are UniFrac-type metrics. However, UniFrac is unsuitable as a distance metric for studies with a small sample size, which is usually the case for dietary intervention trials”.

3. There are many places that authors commented on correlation between taxa, but they did not discuss about in their five real datasets, how correlated the taxa are? Did they on purposely choose the ones contain different level of correlation to distinguish results?

We are not sure what the Reviewer meant here. We argue that multivariate methods should be applied for microbiome data because different OTUs (and corresponding microbial taxa) will be correlated. We did not choose specific taxa based on correlation structure but used filtering criteria related to abundance to avoid extreme sparsity of the data.

4. Bioinformatic pipeline and data quality that used to pre-process can make differences for the downstream analysis for microbiome studies. Any suggestions on statistical tools which can be better/worse in these situations?

We agree with the Reviewer that the bioinformatic pipeline and data quality criteria used can affect the downstream analyses, but this was beyond the scope of our paper.

---

## [Decision Letter · Decision Letter 1]

2 Nov 2021

Analysing microbiome intervention design studies: Comparison of alternative multivariate statistical methods

PONE-D-21-07978R1

Dear Dr. Khomich,

We’re pleased to inform you that your manuscript has been judged scientifically suitable for publication and will be formally accepted for publication once it meets all outstanding technical requirements.

Kind regards,

Lingling An

Academic Editor

PLOS ONE

Additional Editor Comments (optional):

Reviewers' comments:

Reviewer's Responses to Questions

**Comments to the Author**

1. If the authors have adequately addressed your comments raised in a previous round of review and you feel that this manuscript is now acceptable for publication, you may indicate that here to bypass the “Comments to the Author” section, enter your conflict of interest statement in the “Confidential to Editor” section, and submit your "Accept" recommendation.

Reviewer #1: All comments have been addressed

Reviewer #2: All comments have been addressed

2. Is the manuscript technically sound, and do the data support the conclusions?

Reviewer #1: Yes

Reviewer #2: Yes

3. Has the statistical analysis been performed appropriately and rigorously? 

Reviewer #1: Yes

Reviewer #2: Yes

4. Have the authors made all data underlying the findings in their manuscript fully available?

Reviewer #1: Yes

Reviewer #2: Yes

5. Is the manuscript presented in an intelligible fashion and written in standard English?

Reviewer #1: Yes

Reviewer #2: Yes

6. Review Comments to the Author

Reviewer #1: The authors carefully revised the manuscript and address all of my points. I recommend its publication.

Reviewer #2: I'm satisfied with the authors' revisions. A few minor comments on Figures:

1. The legend for Fig. 2 could be more detailed - it took me a while to understand that the numbers indicate percentage of simulated datasets where p-value was significant. Also the lighter colors for "Few_Low" and "Many_Low" scenarios made them a bit difficult to read. I understand they were designed to be consistent with Fig. 3 and 4 colorings. Maybe the authors could make them just a bit darker.

2. Figure 5: I would guess the red points in panels A and C indicate features that are differentially abundant through simulation. If so, the figure legend should clearly indicate this.

7. PLOS authors have the option to publish the peer review history of their article (what does this mean?). If published, this will include your full peer review and any attached files.

Reviewer #1: No

Reviewer #2: No

---

## [Editor Report · Acceptance letter]

8 Nov 2021

PONE-D-21-07978R1 

Analysing microbiome intervention design studies: Comparison of alternative multivariate statistical methods 

Dear Dr. Khomich:

I'm pleased to inform you that your manuscript has been deemed suitable for publication in PLOS ONE. Congratulations! Your manuscript is now with our production department. 

Kind regards, 

on behalf of

Dr. Lingling An 

Academic Editor

PLOS ONE